# Identification of a Novel Variant in *EARS2* Associated with a Severe Clinical Phenotype Expands the Clinical Spectrum of LTBL

**DOI:** 10.3390/genes11091028

**Published:** 2020-09-02

**Authors:** Sofia Barbosa-Gouveia, Emiliano González-Vioque, Álvaro Hermida, María Unceta Suarez, María Jesús Martínez-González, Filipa Borges, Liesbeth Wintjes, Antonia Kappen, Richard Rodenburg, María-Luz Couce

**Affiliations:** 1Diagnosis and Treatment of Congenital Metabolic Diseases Unit (UDyTEMC), Department of Pediatrics, Faculty of Medicine, Clinical University Hospital of Santiago de Compostela, University of Santiago de Compostela, CIBERER, MetabERN, Institute of Clinical Research of Santiago de Compostela (IDIS), 15706 Santiago de Compostela, Spain; egvioque@gmail.com (E.G.-V.); alvaro.hermida@usc.es (Á.H.); lipaborges10@gmail.com (F.B.); maria.luz.couce.pico@sergas.es (M.-L.C.); 2Biochemistry Laboratory (Metabolic Diseases Unit) & Department of Pediatrics (Pediatric Neurology), Cruces University Hospital, 48903 Bizkaia, Spain; maria.uncetasuarez@osakidetza.eus (M.U.S.); mariajesus.martinezgonzalez@osakidetza.eus (M.J.M.-G.); 3Translational Metabolic Laboratory, Department of Laboratory Medicine, Radboud University Medical Centre, 6525 GA Nijmegen, The Netherlands; liesbeth.Wintjes@radboudumc.nl (L.W.); antonia.vanheck-kappen@radboudumc.nl (A.K.); 4Department of Pediatrics, Radboud Centre for Mitochondrial Medicine, Radboud University Medical Centre, 6525 GA Nijmegen, The Netherlands; richard.Rodenburg@radboudumc.nl

**Keywords:** mitochondrial disorders, aminoacyl-tRNA synthetases, *EARS2*, LTBL

## Abstract

The *EARS2* nuclear gene encodes mitochondrial glutamyl-tRNA synthetase, a member of the class I family of aminoacyl-tRNA synthetases (aaRSs) that plays a crucial role in mitochondrial protein biosynthesis by catalyzing the charging of glutamate to mitochondrial tRNA(Glu). Pathogenic *EARS2* variants have been associated with a rare mitochondrial disorder known as leukoencephalopathy with thalamus and brainstem involvement and high lactate (LTBL). The targeted sequencing of 150 nuclear genes encoding respiratory chain complex subunits and proteins implicated in the oxidative phosphorylation (OXPHOS) function was performed. The oxygen consumption rate (OCR), and the extracellular acidification rate (ECAR), were measured. The enzymatic activities of Complexes I-V were analyzed spectrophotometrically. We describe a patient carrying two heterozygous *EARS2* variants, c.376C>T (p.Gln126*) and c.670G>A (p.Gly224Ser), with infantile-onset disease and a severe clinical presentation. We demonstrate a clear defect in mitochondrial function in the patient’s fibroblasts, suggesting the molecular mechanism underlying the pathogenicity of these *EARS2* variants. Experimental validation using patient-derived fibroblasts allowed an accurate characterization of the disease-causing variants, and by comparing our patient’s clinical presentation with that of previously reported cases, new clinical and radiological features of LTBL were identified, expanding the clinical spectrum of this disease.

## 1. Introduction

Mitochondrial diseases are a group of clinically heterogeneous disorders caused by dysfunction of the oxidative phosphorylation (OXPHOS) system and can arise due to defects in genes in the nuclear or mitochondrial DNA (nDNA and mtDNA, respectively) [1]. Human mtDNA contains 37 genes encoding 13 subunits of the OXPHOS complexes I, III, IV, and V and two ribosomal RNAs (rRNA) and 22 transfer RNAs (tRNAs) crucial for their translation [2]. The remaining components of mtDNA replication, transcription, and translation processes, including tRNA maturation, initiation and elongation factors, ribosomal proteins, and aminoacyl-tRNA synthetases (mt-aaRSs), are encoded by nDNA, synthesized in the cytosol, and imported into the mitochondria [3,4]. nDNA-encoded mt-aaRS genes produce enzymes that play an important role in protein synthesis by charging tRNAs with their cognate amino acid and ensuring the correct protein translation [5,6]. Pathogenic variants in mt-aaRSs can directly affect the central nervous system (CNS), leading to characteristic brain lesion patterns (e.g., leukodystrophies) that can be identified by magnetic resonance imaging (MRI) [4,7].

The nuclear gene *EARS2* (MIM*612799) encodes glutamyl-tRNA synthetase, a member of the class I family of mt-aaRSs. This 523-amino acid protein has an N-terminal mitochondrial targeting signal that enables its importation from the cytoplasm into the mitochondria, where it catalyzes the acylation of tRNAGln with glutamine [8,9]. Genetic defects in *EARS2* may lead to combined OXPHOS deficiency (MIM#614924), an autosomal recessive neurologic disorder caused by either compound heterozygous or homozygous variants. Defects in *EARS2* have been associated with a specific clinical syndrome called leukoencephalopathy with thalamus and brainstem involvement and high lactate (LTBL) [10].

LTBL has a broad clinical spectrum, ranging from infantile-onset disease (usually after six months of age) with relatively mild neurological symptoms, followed by spontaneous clinical, biochemical, and radiological improvement, to a more severe phenotype with neonatal/early-infantile onset and rapidly progressive CNS disease that stabilizes but does not improve [10,11,12]. Here, we demonstrate the pathogenicity of *EARS2* variants found in compound heterozygosity in a patient with a variation of the classical LTBL phenotype, characterized by infantile-onset, with progression to a severe clinical phenotype and cerebellar atrophy. We discuss the importance of carefully interpreting variant frequency from population databases and of experimental validation using in-vitro analysis for the accurate characterization of disease-causing variants.

## 2. Materials and Methods

### 2.1. Study Design

This study was carried out in collaboration with the University Clinical Hospital of Santiago de Compostela (Santiago de Compostela, Spain) and Radboud University Medical Center (Nijmegen, The Netherlands). The patient’s parents provided written informed consent to participation in the study and publication of the results. The relevant clinical information from the medical record was collected for the interpretation of the genetic results. For its execution, this project was submitted for approval by the Autonomous Committee of Ethics of Clinical Research of Galicia (code 2015/410). All experimental protocols were approved by the Radboud University Medical Center and were performed in accordance with relevant guidelines and regulations.

### 2.2. Clinical Profile

Our female Spanish patient, the first child of healthy nonconsanguineous parents, was born at 42 weeks of gestation after an uneventful pregnancy. Birth weight, length, and head circumference were all within the normal range. Beginning at 12 months of age, the patient presented daily episodes of absence seizures with mild spasticity of the extremities, lateral pulling of the mouth to the right, and sialorrhea. Electroencephalogram readings were initially normal, although results at 19 years old showed irregular and slightly slowed brain activity for her age. Physical examination revealed microcephaly. The patient required assistance to remain seated, which is her best motor performance achieved. Hepatomegaly was absent. Laboratory analyses of blood, urine, and cerebrospinal fluid (CSF) revealed elevated serum and CSF lactate but no alterations in creatine kinase, glucose, liver function, or blood amino acid levels. Brain magnetic resonance imaging (MRI) revealed generalized leukodystrophy that predominantly affected the thalamus and complete agenesis of the corpus callosum. Assessment of OXPHOS complex enzymatic activity in skeletal muscle at 1 years old revealed a combined deficiency of complexes I and III. Histochemical studies showed normal cytochrome C oxidase (COX) activity, with uniform caliber of the fibers, without structural alterations and no ragged-red fibers. Patient’s mtDNA was sequenced, and the presence of pathogenic variants were discarded. Absence seizures were well-managed at first with empirical phenobarbital treatment. With the appearance of recurrent episodes, carbamazepine was added to the patient’s regimen and was subsequently substituted with vigabatrin. The patient showed progressive intellectual and neurological deterioration and, at 6 years of age, exhibited severe delayed psychomotor development, absence of speech, and spastic tetraparesis. Growth remained poor, with head circumference in the 3rd percentile. Serum lactate levels normalized, and brain MRIs at the ages of 10 and 15 years revealed complete agenesis of the corpus callosum, severe atrophy of the cerebral white matter, and diffuse cerebellar atrophy. By 17 years of age, complete seizure control had not been achieved, and levetiracetam treatment was initiated in parallel with vigabatrin treatment. At follow-up at 19 years, the patient was experiencing 2 seizures episodes per month and was slightly more alert and responsive, although absence of speech and spastic tetraparesis persisted.

### 2.3. Targeted Next-Generation Sequencing

DNA was isolated from the patient’s lymphocytes and analyzed with targeted next-generation sequencing (NGS) panels for mitochondrial diseases. In our unit, we have designed a multi-gene panel consisting of 150 nuclear genes coding for respiratory chain complex subunits and proteins previously implicated in OXPHOS function.

Genetic data were analyzed by NGS technology through a process consisting of enrichment with an in-solution hybridization technology (Sure Select XT; Agilent Technologies, Santa Clara, CA, USA), followed by subsequent sequencing on a Miseq platform (Illumina, San Diego, CA, USA). A custom Sure Select probe library was designed to capture the exons and exon-intron boundaries of the targeted genes [13]. Sequence capture, enrichment, and elution were performed in accordance with the manufacturer’s instructions. Image analysis and processing of the sequence fluorescence intensity (“base calling”) was performed with Real Time Analysis (RTA) software v.1.8.70 (Illumina), and the FastQC v0.10.1 program was used for data quality control. Reads were aligned to the reference genome GRCh37 using BWA v0.7.9a [14]. NGSrich v0.7.5 software [15] was used as a control prior to variant detection and BEDTools 2.17.0 [16] and Picard 1.114 [17] for intermediate steps. VarScan v.2.3.6 [18] and SAMtools v0.1.19 [19] were used for variant detection for indels and Single Nucleotide Polymorphism (SNPs), respectively, and ANNOVAR for variant annotation [20].

To ensure reliable clinical interpretation of the detected variants, prioritization criteria were applied to predict their pathogenicity in accordance with the ACMG assay mix containing 0.3-mM acetyl-CoA guidelines [21].

### 2.4. Cell Culture

Patient fibroblasts from skin biopsy performed at 19 years old and a control fibroblast line routinely used for research purposes were cultured in M199 medium (Gibco, Thermo Fisher Scientific, Waltham, MA, USA) supplemented with 10% *v*/*v* fetal calf serum (FCS) and 1% *v*/*v* penicillin/streptomycin (Gibco) at 37 °C and 5% CO_2_.

### 2.5. Mitochondrial Isolation

Pelleted fibroblasts from the patient and from the control cell line were resuspended in ice-cold Tris-HCl (10 mM, pH 7.6). Cells were then disrupted in a Potter-Elvejhem homogenizer at 1800 rpm, and sucrose (250 mM) was added to make the samples isotonic. The homogenized cell samples were then centrifuged for 10 min at 600× *g*, and the mitochondria pellet was obtained after centrifugation of the supernatant for 10 min at 14,000× *g*.

### 2.6. Respirometry and Measurement of OXPHOS Activity

A Seahorse XFe96 Extracellular Flux analyzer (Seahorse Bioscience, Agilent) was used to measure the oxygen consumption rate (OCR). On the day before the assay, control and patient fibroblasts were seeded at 10,000 cells per well in cell culture medium (M199 supplemented with 10% FCS and 1% pen/strep) and incubated overnight at 37 °C and 5% CO_2_. On the day of the assay, cell culture medium was replaced with Agilent Seahorse XP Base Medium with 10-mM glucose (Sigma Aldrich, St. Louis, MO, USA), 1-mM sodium pyruvate (Gibco), and 200-mM l-glutamine (Life Sciences Group, Barnet, UK) and then incubated for 1 h at 37 °C without CO_2_. Baseline cellular OCR was measured 8 times, followed by 4 measurement cycles after addition of the following inhibitors: 1-µM oligomycin A (Sigma), 2.0-µM and 4.0-µM carbonyl cyanide 4-(trifluoromethoxy) phenylhydrazone (FCCP) (Sigma), and 0.5-µM rotenone and 0.5-µM antimycin A (Sigma). After OCR measurements, the cell medium was removed and replaced with 0.33% Triton X-100, and 10-mM Tris-HCl (pH 7.6). Seahorse plates were stored at −80 °C and thawed afterwards. Citrate synthase activity was measured spectrophotometrically, at 37 °C, using a Tecan Spark spectrophotometer. The assay mix contained 0.3-mM acetyl-CoA, 0.1-mM dithionitrobenzoic acid (DTNB), 0.025% Triton X-100, and 10-mM Tris-HCl, pH 8.1. Measurements were based on the absorption at 412 nm of thionitrobenzoic acid (TNB), and citrate synthase activity was calculated based on the rate of dithionitrobenzoic acid (DTNB) conversion in the presence of oxaloacetate. OCR was measured before and after the addition of inhibitors and was normalized to the activity of the citrate synthase (CS) [22] to avoid the effect of differences in the mitochondrial content or cell number.

The enzymatic activities of complexes I–V were assayed spectrophotometrically using mitochondrial extracts prepared as previously described in mitochondrial isolation. These measurements were performed according to the current protocols in use in the Nijmegen Center for Mitochondrial Disorders [23]. All assays were performed in duplicate using a Konelab 20XT auto-analyzer (Thermo Fischer Scientific).

## 3. Results

### 3.1. Molecular Genetics and In-Silico Analysis of EARS2 Variants

NGS analysis enabled the identification of two compound heterozygous variants in *EARS2* (NM_001083614.1) in our patient (Figure 1): variant c.670G>A (p.Gly224Ser) has been previously described in a patient with mild LTBL [10], while c.376C>T (p.Gln126*) has not to date been included in any human genetic variation database.

The identified *EARS2* variants were analyzed in silico to determine the degree of evolutionary conservation, predicted pathogenicity, functional consequences, and minor allele frequency (MAF) within the population (Appendix A). According to GERP [24], PhyloP [25], and phastCons [25], both variants are located in areas of the *EARS2* protein that are highly conserved through evolution. The pathogenicity was predicted using MutationTaster [26] (prediction and disease-causing) and by calculating the FATHMM [27] and DANN [28] scores, which corresponded to “damaging”. The missense variant c.670G>A is located in exon 4 and causes an amino acid change from a nonpolar glycine to an uncharged polar serine at residue 224. Both SIFT [29] and Provean [30] software predicted a damaging effect of this variant on the protein function. The nonsense variant c.376C>T, located in exon 3, was predicted to be disease-causing and damaging by MutationTaster and FATHMM, respectively.

Segregation studies were performed by Sanger sequencing to determine the inheritance pattern of the identified variants (Figure 1). The missense variant was inherited from the patient’s mother and the nonsense variant from the father.

### 3.2. Mitochondrial Respiration and OXPHOS Activity

Measurement of the OCR and extracellular acidification rate (ECAR), normalized by the mitochondrial content, revealed decreases in both the OCR and the OCR/ECAR ratio in the patient’s fibroblasts, indicating a reduced electron flow through the respiratory chain (Figure 2A,B). These findings reflect a general deficiency in the cumulative proficiency of the entire set of mitochondrial respiratory chain complexes in the patient’s fibroblasts.

The spectrophotometric analysis of respiratory chain enzyme activity revealed a slight deficiency in complex III in the patient’s fibroblasts and normal levels of activity of complexes I, II, IV, and V (Figure 2C). Additionally, mitochondrial mass differences were detected between the patient and control—160 U/L and 339 U/L, respectively—regarding the CS measurement.

## 4. Discussion

LTBL is a condition with a well-defined clinical phenotype, characterized by extensive symmetrical deep white matter abnormalities and signal changes in the thalami and brainstem, along with lactic acidosis [10,31]. Two distinct clinical courses are described: severe and moderate. The severe form is characterized by early-onset, delayed psychomotor development, seizures, hypotonia, and persistently elevated lactate levels, with progressive atrophy of the affected brain structures. The mild form usually manifests clinically after six months of age with irritability and psychomotor regression, although patients can show clinical and biochemical improvements. The occurrence and extent of recovery most likely depends on the severity of brain damage caused by the first symptomatic episode; some children with the most severe phenotype can die after the first clinical event [31,32,33,34]. Less common findings reported in LTBL include anemia, tubulopathy, hepatopathy, absence of the thalami, and even the absence of characteristic neuroimaging findings [10,35,36,37]. The clinical course of our patient does not appear to correspond to either of the two clinical courses described: she has a severe clinical presentation that initially manifested at 12 months of age, characterized by absence seizures with mild spasticity of the extremities. In addition to the typical clinical features of LTBL, MRIs at 10 (Figure 3A) and 15 (Figure 3B) years of age revealed diffuse atrophy of the cerebellum that was not present in the early stages of the disease. Later-onset of cerebellar atrophy was also identified in a patient reported by Steenweg et al. (Appendix A). In addition to the cerebellar atrophy associated with LTBL (Appendix A), our patient exhibited progressive brain lesions with no signs of improvement, which is commonly reported in cases diagnosed after four months of age [10,36,38,39]. It has been difficult to establish a correlation between patients’ genotypic and phenotypic characteristics, since the same variant may be associated with different clinical signs, even among members of the same family [10,39].

The evaluation of all LTBL cases with homozygous or compound heterozygous variants in *EARS2* reported to date indicates that certain aspects of the established LTBL phenotype do not appear in all cases of this pathology. As shown in Appendix A, leukoencephalopathy was absent in patients 14 and 26, the thalamus and brainstem involvement was absent in patients 18 and 29, and lactic acidosis was absent in patient 18 (clinical conditions that do not correspond to the classical LTBL phenotype are shown in bold).

The data presented here support the pathogenicity of the compound heterozygous *EARS2* variants detected in our patient. We provide the first description of the nonsense variant c.376C>T (p.Gln126*), inherited from the patient’s father. Our patient also carried the variant c.670G>A (p.Gly224Ser), which was previously identified in a patient (patient 7, Appendix A) with mild LTBL by Steenweg et al. [10]. Although this variant has a high frequency in allele-population databases such as gnomAd and ExAC (0.00087 and 0.00119, respectively), especially in African populations, a software in-silico analysis predicted a high level of pathogenicity for this variant. The combination of severe changes, like nonsense with milder missense variants, is a common molecular mechanism in *EARS2*-associated diseases.

Mitochondrial function characterization using the patient’s fibroblasts revealed a mitochondrial dysfunction due to abnormal mitochondrial respiration, as evidenced by a decrease in the OCR/ECAR ratio. The decrease in OCR and the increase in lactate production observed in our patient are likely the consequences of deficient respiration due to general impairment of the entire set of mitochondrial respiratory chain complexes. The analysis of respiratory chain enzymatic activity in muscle samples from our patient revealed deficiencies in complexes I and III. However, the same analysis of fibroblast samples only showed a mild decrease in complex III activity. Despite our patient’s severe clinical presentation and the clear deficit in fibroblast mitochondrial function, these discrepancies between muscle and fibroblast findings are not uncommon in nuclear-encoded gene-associated disorders [23]. Importantly, the various *ARS2* gene defects are notoriously variable when it comes to issue-specific biochemical phenotypes [40].

Our findings highlight the complexity of OXPHOS enzymes. On the one hand, we have a system composed of multi-subunit enzymes encoded both by mtDNA and nDNA (except for complex II, which is only encoded by nDNA). Consequently, pathogenic variants in both genomes can lead to the highly heterogeneous clinical presentations characteristics of mitochondrial disorders. On the other hand, this tight relationship guarantees a close coevolution of the two genomes. Therefore, any inconsistency affecting either genome may have severe negative effects on the cellular fitness [41]. In fact, studies of *Drosophila melanogaster* have shown that interactions between nDNA-encoded mt-aaRSs and mitochondrial tRNAs may not only delay development and fecundity but, also, affect the activities of oxidative phosphorylation complexes I, III, and IV, suggesting that these interactions may be targets of compensatory molecular evolution [42]. Several features of mtDNA, including a high mutation rate and uniparental inheritance, makes mtDNA more likely to accumulate mutations in the population. These changes, in turn, could drive adaptations in nDNA, given the intimate interactions between both genomes. In different populations, this coevolution is driven by different factors and in different directions [41,43], leading to distinct consequences of nDNA-encoded variants, depending on the mtDNA background. This could explain why variant c.670G>A (p.Gly224Ser), which was detected in our patient and in the LTBL case described by Steenweg et al., is predicted as pathogenic and associated with LTBL disease despite being classified as benign, owing to its high frequency in African populations. This hypothesis underscores the need for caution when consulting population frequency databases, particularly when studying variants in nuclear genes that interact directly with mitochondrial genes.

Taken together, the results of the functional analyses and family studies presented here enabled the correct identification of the pathogenicity of the variants identified in our patient, thereby expanding the clinical spectrum of LTBL to include cerebellar atrophy. Establishing the pathogenicity has significant implications for the diagnosis and management of these disorders. However, the relationship between *EARS2* variants and the phenotypic characteristics of LTBL remain poorly understood, and further studies are needed to elucidate the molecular mechanisms underlying this disease.

## Figures and Tables

**Figure 1 genes-11-01028-f001:**
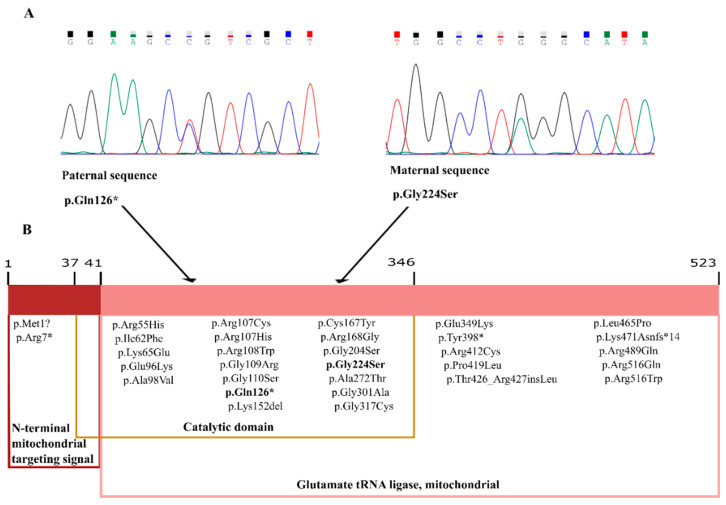
Reverse sequence chromatograms showing the results of Sanger sequencing. (**A**) The missense variant c.670G>A (p.Gly224Ser), located in exon 4, was inherited from the patient’s mother and the nonsense variant c.376C>T (p.Gln126*), located in exon3, from the father. The patient harbored both variants in *EARS2* in compound heterozygosity. (**B**) Schematic showing *EARS2* domains in which all variants identified to date in leukoencephalopathy with thalamus and brainstem involvement and high lactate (LTBL) patients are highlighted.

**Figure 2 genes-11-01028-f002:**
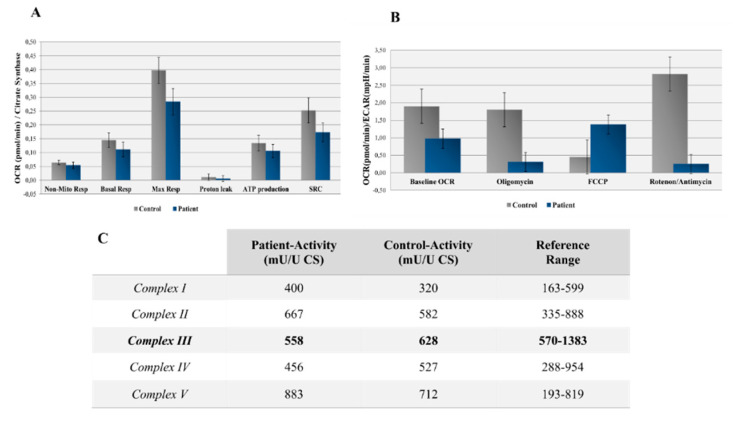
(**A**) Oxygen consumption rate (OCR) measured before and after the addition of inhibitors. The Seahorse XF Cell Mito Stress Test uses compounds of respiration that target components of the electron transport chain (ETC) in the mitochondria to reveal key parameters of the metabolic functions. These modulators are ETC inhibitors (oligomycin, carbonyl cyanide 4-(trifluoromethoxy) phenylhydrazone (FCCP), and a mixture of rotenone and antimycin A), which are serially injected to measure the ATP production, maximal respiration (Max Resp), nonmitochondrial respiration (Non-Mito Resp), proton leak, spare respiratory capacity (SRC), and basal respiration (Basal Resp). (**B**) The basal energy metabolism of each cell line was assessed by analyzing the OCR/extracellular acidification rate (ECAR) ratios following the sequential injection of the inhibitors. (**C**) Measurements of the enzyme activities for the different oxidative phosphorylation (OXPHOS) complexes in the patient’s fibroblasts (reference range deemed in a healthy population). CS, citrate synthase.

**Figure 3 genes-11-01028-f003:**
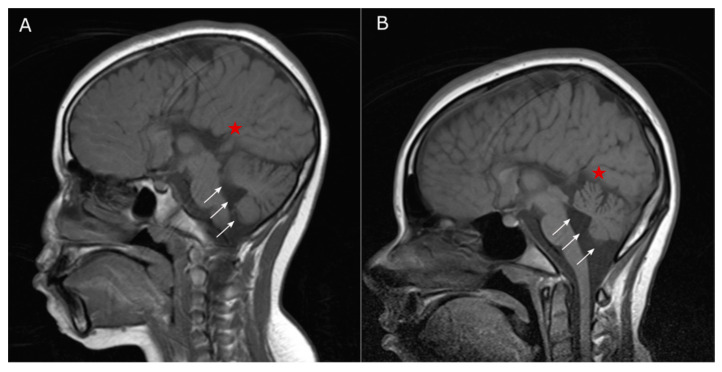
MRI images of our patient acquired at 10 (**A**) and 15 (**B**) years of age, showing complete agenesis of the corpus callosum (indicated with a red star) and cerebellar atrophy (indicated with white arrows). Note also the dilation of the fourth ventricle and enlargement of the interfolial spaces within a normal-sized posterior fossa.

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
