# Peer review of "Identification of a Novel Variant in EARS2 Associated with a Severe Clinical Phenotype Expands the Clinical Spectrum of LTBL"

_genes, 2020, doi:10.3390/genes11091028_

Round 1
Reviewer 1 Report
The report "Identification of a novel variant in EARS2 associated with a severe clinical phenotype expands the clinical spectrum of LTBL" by Barbosa-Gouveia et al identified a compound heterozygous EARS2 variants as the likely source of LTBL pathology in a patient using NGS targeting a panel of 150 nuclear genes. One of the EARS variants identified has never been reported before while the other variant was reported to show high frequency in African populations. The case study was clearly presented and documents a new deleterious mutation variant for EARS, and how it is linked to mitochondria dysfunction.
Comments and suggestions:
1. While I understand using a set of targeted panel, the authors should indicate why they excluded analyzing the mtDNA from the patient, which can contribute/explain the variation in the clinical presentations.
2. Does Gln126* lead to nonsense decay of the associated mRNA or does it lead to a truncated protein product?
3. Do data in Figure 2a and 2b show statistical significance? Please include all statistical analysis.
4. The measurement of OCAR are not always normalized by CS, although it is fine. The authors should be clear in their interpretation of what it means when the data are normalized this way, i.e. OxPhos activity per mitochondria, which does not necessarily correlate to OxPhos activity per cell.
5. The most troubling aspect of the report is the lack of direct verification of suspected mitochondrial translation deficiency. The authors should demonstrate with western blot analysis or immunofluorescence analysis that the compound heterozygous EARS2 variants reported here lead a breakdown of OxPhos complexes. Alternatively, the mitochondria translational rate should be measured to demonstrate the casual relationship.
6. It would be fairly easy to examine how the state of OxPhos complex assembly might impact the health of mitochondria population from the patient fibroblasts. Are mitochondria mass and morphology altered? Again, simple staining of mitotracker would answer this question. Also, are the ROS burden of these cells altered, which can be measured using specific dye? If the mitochondria becomes more depleted due to the reported EARS2 variants, then one can expect the mitochondria dysfunction to be even more enhanced when compared on the cellular level.
7. For the Seahorse measurements, what control is used? This is not clearly described. Although it might be difficult, it would be interesting to compare to the samples from the parents, especially the novel variant, which presumably has only subtle effects.
8. How were the measurements in Fig 2c carried out? Is this done with purified mitochondria or mitochondria extracts? It is unclear why the method section listed the method to purify mitochondria. The reference cited offered a range of methods and the author should explicitly explain which methods were used. Or the authors should state what company manufactured the kit if a kit were used. The data from this experiments was indeed surprising. Is there compensating mechanism at work here?
9. Line 267, is "ARS2" gene a typo?
Author Response
We thank the reviewers for their positive comments and suggestions that have certainly helped to increase the overall quality of the manuscript. All the additions to the manuscript are highlighted in yellow in the revised version.
Comments and suggestions:
- While I understand using a set of targeted panel, the authors should indicate why they excluded analyzing the mtDNA from the patient, which can contribute/explain the variation in the clinical presentations.
The presence of pathogenic variants in the patient´s mtDNA had been previously discarded, although that analysis was not performed in our lab. We agree with the reviewer that mtDNA studies should be performed combined with mitochondrial nuclear genes sequencing for mitochondrial disorders diagnosis. We have included a sentence in the revised manuscript (lines 100-101 of Clinical Profile in the Experimental Section) to clarify this point.
- Does Gln126* lead to nonsense decay of the associated mRNA or does it lead to a truncated protein product?
Although we agree with the reviewer that this is an interesting question, we think these experiments could be beyond the scope of this study. EARS2 nonsense variants have been previously associated with LTBL and the combination of predictably severe change, like nonsense, with milder mutations, like missense mutation in moderately conserved domains, as the combination observed in our patient, is a common molecular mechanism in EARS2 associated diseases. So we think that from a clinical point of view the results are robust enough to establish a diagnostic.
- Do data in Figure 2a and 2b show statistical significance? Please include all statistical analysis.
Mitochondrial stress tests performed with Seahorse XF Analyzers allow the estimation of different bioenergetic measures by monitoring the oxygen consumption rates of fibroblasts in multi-well plates in the same experiment. Normalization of the resultant data minimizes inconsistency and variations from well-to-well, average, and the standard deviation is then calculated. Statistical analysis is not usually performed in these tests. Only the Grubbs’s test is applied for the identification of outliers.
- The measurement of OCAR are not always normalized by CS, although it is fine. The authors should be clear in their interpretation of what it means when the data are normalized this way, i.e. OxPhos activity per mitochondria, which does not necessarily correlate to OxPhos activity per cell.
Oxygen Consumption Rate (OCR) is already normalized by mitochondrial mass using CS activity (Figure 2A).
- The most troubling aspect of the report is the lack of direct verification of suspected mitochondrial translation deficiency. The authors should demonstrate with western blot analysis or immunofluorescence analysis that the compound heterozygous EARS2 variants reported here lead a breakdown of OxPhos complexes. Alternatively, the mitochondria translational rate should be measured to demonstrate the casual relationship.
We agree with the reviewer that to demonstrate the functional link between variants and phenotypes should be the goal when describing new variants or new genes. On the other hand, the main aim of our work was to describe a new patient with a LTBL plus phenotype associated with EARS2 variants. The genotype of the patient, an already described missense variant, and a new nonsense variant, fit perfectly with the molecular bases of the disease. As we have answered before, the combination of severe change, like nonsense, with milder missense mutations is a common molecular mechanism in EARS2 associated diseases. So we think that from a clinical point of view the results are robust enough to establish a genetic diagnostic.
- It would be fairly easy to examine how the state of OxPhos complex assembly might impact the health of mitochondria population from the patient fibroblasts. Are mitochondria mass and morphology altered? Again, simple staining of mitotracker would answer this question. Also, are the ROS burden of these cells altered, which can be measured using specific dye? If the mitochondria become more depleted due to the reported EARS2 variants, then one can expect the mitochondria dysfunction to be even more enhanced when compared on the cellular level.
We have not data about mitochondrial mass or morphology aside of the CS activity, which showed differences between patient and control fibroblasts. We think that the OXPHOS dysfunction of the patient is clearly demonstrated both at muscle and cellular (fibroblast) level. Assessment of OXPHOS complex enzymatic activity in skeletal muscle revealed a combined deficiency of complexes I + III, and the analysis using fibroblast cell line showed a clear functional defect of the OXPHOS system (Figure 2). Mitotracker will show differences more due to the expected defective OXPHOS function and its effect on mitochondrial membrane potential than mitochondrial mass or morphology alterations. Although the study of the implication of the ROS metabolism in the pathogenesis of EARS2 related disorders would be of interest, it is beyond the scope of this study.
- For the Seahorse measurements, what control is used? This is not clearly described. Although it might be difficult, it would be interesting to compare to the samples from the parents, especially the novel variant, which presumably has only subtle effects.
The control used in Seahorse measurements was a fibroblast cell line routinely used in the Translational Metabolic Laboratory (Radboud University Medical Centre) for this porpoise. We agree that it would be interesting to compare the mitochondrial function of the patient´s fibroblast with those from the parents. Unfortunately, those samples weren´t available.
- How were the measurements in Fig 2c carried out? Is this done with purified mitochondria or mitochondria extracts? It is unclear why the method section listed the method to purify mitochondria. The reference cited offered a range of methods and the author should explicitly explain which methods were used. Or the authors should state what company manufactured the kit if a kit were used. The data from this experiments was indeed surprising. Is there compensating mechanism at work here?
The OXPHOS complexes measurements were carried out using mitochondrial extracts prepared as described in lines 143-147 (mitochondrial isolation) from the Experimental Section.
Differences between OXPHOS enzyme activities in muscle and fibroblast are common for mitochondrial disorders caused by pathogenic variants in nuclear genes. For mutations in aminoacyl-tRNA synthetase gene, normal or mild OXPHOS deficiencies in fibroblast, like that found in the patient, have been frequently reported. Postmitotic tissues, like muscle, are more sensible to those defects, while proliferative ones, like fibroblast cell lines, usually don´t express the deficiencies.
According to reference 23 regarding the method for enzymatic activities: “Biochemical diagnostic examination of tissue and cell samples from mitochondrial patients includes measurements of enzyme activities of the oxidative phosphorylation (OXPHOS) system, consisting of complex I (EC 1.6.5.3), complex II (EC 1.3.5.1), complex III (EC 1.10.2.2), complex IV (EC 1.9.3.1), and complex V (EC 3.6.1.3) (Fig. 1). Assays to quantify OXPHOS enzyme activities are usually based on spectrophotometry (Benit et al. 2006; Janssen et al. 2007; Kirby et al. 2007; Rustin et al. 1994)”
- Line 267, is "ARS2" gene a typo?
No, it is ok. This part of the discussion compares the cellular phenotypes associated with defects in aminoacyl-tRNA synthetases.
Reviewer 2 Report
Identification of a novel variant in EARS2 associated with a severe clinical phenotype expands the clinical spectrum of LTBL
Sofia Barbosa-Gouveia et al. 2020
Using targeted sequencing the authors identified two candidate pathogenic variants in EARS2 encoding the mitochondrial glutamyl-tRNA synthetase, that has important roles in mitochondrial protein synthesis, as it catalyzes the charging of glutamate to mitochondrial tRNA(Glu). The authors nicely summarize the previous studies around EARS2 variants in a supplementary table, providing a detailed table of clinical features which are predominantly associated with leukoencephalopathy with thalamus and brainstem involvement and high lactate (LTBL). The patient identified in this study was carrying two heterozygous EARS2 variants; a newly identified c.376C>T (p.Gln126*) and a previously characterized c.670G>A (p.Gly224Ser). The patient presented with similar clinical phenotypes associated with the severe cases of LTBL, with some additional clinical and radiological features, that may have not been sufficiently highlighted in previous studies. The authors performed segregation studies and provide biochemical data (oxygen consumption rates, respiratory chain enzyme activity measurements) to demonstrate an effect of the variants on mitochondrial function. The study expands the group of patients harbouring EARS2 variants and identifies a novel variant in EARS2, which will be of interest. I would like the authors to address the general comments and the more specific points below.
General comments:
- Certain statements in the discussion and abstract are rather strong and I think the authors should down-tune this. The overall clinical phenotype of the patient does seem to fit the general clinical description of patients carrying EARS2 mutations. However, there is no direct proof of pathogenicity/or underlying molecular mechanism for the EARS2 variants identified in this study - this would potentially include further functional work up such as determining EARS2 protein levels, complementation/rescue studies or even aminoacylation studies. Although, I believe these experiments may be beyond the scope of this study, I think the authors should point this out and also the fact that a targeted gene panel for 150 genes has been used, rather than WES or WGS and therefore it cannot be excluded that other candidate gene variants may contribute to the disease phenotype.
- The authors report new clinical and radiological features of LTBL, that includes cerebellar atrophy that was not present at earlier stages of the disease, but it appears at later age. The authors should revisit the Steenweg et al. 2012 Brain paper [10] describing a subset of patients carrying EARS2 mutations. The detailed supplementary Table 3 in this study suggests similar clinical phenotype related to the cerebellar atrophy described by the authors - where the two severe LTBL cases (Patient 9 and Patient 10) have no signs of atrophy at early infancy/childhood, however on follow up MRI studies the abnormalities appear. The authors should include these findings.
- The discussion around the c.670G>A (p.Gly224Ser) - this mutation is predicted as pathogenic in the Steenweg et al. and this study using in silico prediction tools, based on conservation, amino substitution etc. however, the MAF appears high (in African population). I’ll be interested what was the ethnicity of the patient in this study – was the patient of African origin? The one in Steenweg’s study appears to be from Switzerland, but it is not known what ethnicity the patient was. This could have an impact on the interpretation of the MAF?
- The authors should discuss that potential affects of the p.Glu126* mutation – haploinsufficiency? As this mutation is introduced relatively early on within the EARS2 sequence, the truncated gene product is likely to undergo nonsense mediated mRNA decay.
Specific points:
Abstract line 36 – ‘’…we describe the molecular mechanism underlying the pathogenicity of these EARS2 ‘’ Please reword this sentence as the paper identifies candidate pathogenic variants in EARS2 but does not describe any particular molecular mechanism that underly EARS2 pathogenicity.
Abstract line 37 – please remove ‘in vitro analysis’ and use e.g. patient-derived fibroblasts and muscle tissue
Introduction line 53 – Reference [3] Christian, B.E.; Spremulli, L.L. Mechanism of Protein Biosynthesis in Mammalian Mitochondria. Biochim Biophys Acta 2012, is rather out of date and does not provide a review of all the processes described in the sentence. Please update the references. Some suggestions: PMID 30030363 that covers most of transcription/translation and PMID 29880722 for mt replication.
Introduction line 57 – please also add this recent reference that focuses on EARS2: PMID 31839000 Mitochondrial aminoacyl-tRNA synthetase disorders: an emerging group of developmental disorders of myelination, JND, 2019.
Introduction line 58 – the authors should provide the NCBI NM_ accession number for EARS2
Introduction line 62 – ‘’Genetic defects in EARS2 may lead to combined OXPHOS deficiency (MIM#614924)’’ - Could the authors provide the appropriate references for this or refer to the supplementary Table 2.
Experimental section line 114 – authors should provide the appropriate reference for the gene panel if previously published or provide a supplementary table of the examined genes.
Experimental section line 133 – It appears that only one control was used in the study. Was the control age-matched? Also, it is not clear whether the patient undergone skin biopsy at infancy or early adulthood (19 years old at last follow up?). Authors should clarify these points for the experimental procedures that involve fibroblast studies and also should indicate whether the muscle biopsy used for the enzyme activity measurements was taken at similar times as the fibroblast biopsy?
Results line 176, 179, 181 – authors should add the appropriate references or web-links for the prediction tools used in the study.
Results line 189 – Please delete the additional ‘c’ in cc.670G>A
Results lines 194-196 – I find the description of the results from the OCR and ECAR rates insufficient and the authors need to rewrite this section and provide more details of the results and discuss their findings.
Figure 1a – Parental sequencing chromatographs are shown but the sequencing for the proband is not shown. Can the authors provide the sequencing analysis for the affected patient?
Figure 2a – there is a spelling mistake in the y-axis for citrate synthase
Figure 2 – can the authors ‘tidy up’ the figures e.g. remove additional lines and align text so it’s not below the lines (figure 2a).
Author Response
We thank the reviewers for their positive comments and suggestions that have certainly helped to increase the overall quality of the manuscript. All the additions to the manuscript are highlighted in yellow in the revised version.
General comments:
1. Certain statements in the discussion and abstract are rather strong and I think the authors should down-tune this. The overall clinical phenotype of the patient does seem to fit the general clinical description of patients carrying EARS2 mutations. However, there is no direct proof of pathogenicity/or underlying molecular mechanism for the EARS2 variants identified in this study - this would potentially include further functional work up such as determining EARS2 protein levels, complementation/rescue studies or even aminoacylation studies. Although, I believe these experiments may be beyond the scope of this study, I think the authors should point this out and also the fact that a targeted gene panel for 150 genes has been used, rather than WES or WGS and therefore it cannot be excluded that other candidate gene variants may contribute to the disease phenotype.
Following reviewer suggestions, we have changed the sentence in lines 35-37 of the abstract for:
“We demonstrate a clear defect in mitochondrial function in the patient’s fibroblasts suggesting the molecular mechanism underlying the pathogenicity of these EARS2 variants”
and sentence in 261-263 lines of discussion for:
“Mitochondrial function characterization using the patient’s fibroblasts revealed a mitochondrial dysfunction due to abnormal mitochondrial respiration, as evidenced by a decrease in the OCR/ECAR ratio”
Although we agree with the reviewer that the results presented in our manuscript didn´t demonstrate without doubt the underlying molecular mechanism, we think that from a clinical point of view the results are robust enough to establish a genetic diagnostic. The genotype of the patient, an already described missense variant, and a new nonsense variant, fit perfectly with the molecular bases of the disease. The combination of severe change, like nonsense, with milder missense mutations is a common molecular mechanism in EARS2 associated diseases, and, like the reviewer point out, the clinical phenotype of the patient fits the clinical phenotype of patients carrying EARS2 mutations.
At the moment this patient was studied, the diagnostic tool used in our lab was the targeted panel, and the mitochondrial panel was composed of those 150 genes. Because we are mainly a clinical lab, the first-tier genetic test we used is usually focused on the analysis of exonic or intronic variants in known disease-associated genes or variants with demonstrated pathogenicity located in noncoding regions, while WES or WGS approaches are only used when the routine test failed to establish a genetic diagnostic. In our opinion, the 150 genes panel applied to this patient was useful to establish a genetic diagnostic in this case.
2. The authors report new clinical and radiological features of LTBL, that includes cerebellar atrophy that was not present at earlier stages of the disease, but it appears at later age. The authors should revisit the Steenweg et al. 2012 Brain paper [10] describing a subset of patients carrying EARS2 mutations. The detailed supplementary Table 3 in this study suggests similar clinical phenotype related to the cerebellar atrophy described by the authors - where the two severe LTBL cases (Patient 9 and Patient 10) have no signs of atrophy at early infancy/childhood, however on follow up MRI studies the abnormalities appear. The authors should include these findings.
Following reviewer suggestions, we have reviewed Steenweg et al. manuscript and indeed patient 9 had developed cerebellar atrophy at 4 years-old. This information was added to the manuscript (lines 233-235 of discussion). Although, in our opinion, the severe clinical presentation of our case is not common since later-onset was associated until now to a moderate course of the disease.
“Later-onset of cerebellar atrophy was also identified in a patient reported by Steenweg et al. (Table 2, Supplementary material).”
3. The discussion around the c.670G>A (p.Gly224Ser) - this mutation is predicted as pathogenic in the Steenweg et al. and this study using in silico prediction tools, based on conservation, amino substitution etc. however, the MAF appears high (in African population). I’ll be interested what was the ethnicity of the patient in this study – was the patient of African origin? The one in Steenweg’s study appears to be from Switzerland, but it is not known what ethnicity the patient was. This could have an impact on the interpretation of the MAF?
The patient described in our manuscript is from Spanish origin. As we discussed in the manuscript (lines 255-257 of discussion) although this variant has a high frequency in allele-population databases such as gnomAd and ExAC (0.00087 and 0.00119, respectively), especially in African populations, software in silico analysis predicted a high level of pathogenicity for this variant. The frequency of the allele in European (Non-Finnish) population is 0.000124, much lower than in African population. We thought that the interpretation of the MAF need to be done taking into consideration this difference, especially, as we pointed out in the discussion section when studying variants in nuclear genes that interact directly with mitochondrial genes. Following reviewer suggestion, Spanish ethnicity was added to the clinical profile.
4. The authors should discuss that potential affects of the p.Glu126* mutation – haploinsufficiency? As this mutation is introduced relatively early on within the EARS2 sequence, the truncated gene product is likely to undergo nonsense mediated mRNA decay.
The combination of severe change, like nonsense, with milder missense mutations is a common molecular mechanism in EARS2 associated diseases. The c.1194C>G (p.Tyr398Ter) variant described in Steenweg’s study is predicted to cause loss of normal protein function through protein truncation, and in ClinVar database are described another three nonsense/frameshift pathogenic variants: c.684C>A (p.Tyr228Ter); c.212del (p.Phe71SerfsX3); and c.1283del (p.Pro428fs). There aren´t, to the best of our knowledge, functional studies of the molecular mechanism of these variants which lead to loss of function of EARS2 protein caused by nonsense/frameshift variants. We have added this sentence (lines 257-259 of discussion):
“The combination of severe change, like nonsense, with milder missense mutations is a common molecular mechanism in EARS2 associated diseases”
Specific points:
Abstract line 36 – ‘’…we describe the molecular mechanism underlying the pathogenicity of these EARS2 ‘’ Please reword this sentence as the paper identifies candidate pathogenic variants in EARS2 but does not describe any particular molecular mechanism that underly EARS2 pathogenicity.
Following reviewer suggestions, we have change the sentence in lines 35-37 of the abstract for:
“We demonstrate a clear defect in mitochondrial function in the patient’s fibroblasts suggesting the molecular mechanism underlying the pathogenicity of these EARS2 variants”
Abstract line 37 – please remove ‘in vitro analysis’ and use e.g. patient-derived fibroblasts and muscle tissue
Following reviewer suggestions, we have change the sentence in line 37 to:
“Experimental validation using patient-derived fibroblasts”
Introduction line 53 – Reference [3] Christian, B.E.; Spremulli, L.L. Mechanism of Protein Biosynthesis in Mammalian Mitochondria. Biochim Biophys Acta 2012, is rather out of date and does not provide a review of all the processes described in the sentence. Please update the references. Some suggestions: PMID 30030363 that covers most of transcription/translation and PMID 29880722 for mt replication.
Following reviewer suggestions, we have change the reference number 3 to:
D'Souza AR, Minczuk M. Mitochondrial transcription and translation: overview. Essays Biochem. 2018, 62(3), 309-320
Introduction line 57 – please also add this recent reference that focuses on EARS2: PMID 31839000 Mitochondrial aminoacyl-tRNA synthetase disorders: an emerging group of developmental disorders of myelination, JND, 2019.
Following reviewer suggestions, we have change the reference number 7 to:
Fine AS, Nemeth CL, Kaufman ML, Fatemi A. Mitochondrial aminoacyl-tRNA synthetase disorders: an emerging group of developmental disorders of myelination. J Neurodev Disord. 2019;11(1):29.
Introduction line 58 – the authors should provide the NCBI NM_ accession number for EARS2
The accession number for the EARS2 transcript used (NM_001083614.1) is provided in the Results section, line 175
Introduction line 62 – ‘’Genetic defects in EARS2 may lead to combined OXPHOS deficiency (MIM#614924)’’ - Could the authors provide the appropriate references for this or refer to the supplementary Table 2.
The reference for the EARS2 asociated phenotype is provided in line 65:
“Defects in EARS2 have been associated with a specific clinical syndrome called leukoencephalopathy with thalamus and brainstem involvement and high lactate (LTBL) [10].”
- Steenweg, M.E.; Ghezzi, D.; Haack, T.; Abbink, T.E.M.; Martinelli, D.; van Berkel, C.G.M.; Bley, A.; Diogo, L.; Grillo, E.; Te Water Naudé, J.; et al. Leukoencephalopathy with thalamus and brainstem involvement and high lactate “LTBL” caused by EARS2 mutations. Brain 2012, 135, 1387–1394.
Experimental section line 114 – authors should provide the appropriate reference for the gene panel if previously published or provide a supplementary table of the examined genes.
Following reviewer suggestions, we have added a supplementary table (Table S3) with the list of genes examined.
Experimental section line 133 – It appears that only one control was used in the study. Was the control age-matched? Also, it is not clear whether the patient undergone skin biopsy at infancy or early adulthood (19 years old at last follow up?). Authors should clarify these points for the experimental procedures that involve fibroblast studies and also should indicate whether the muscle biopsy used for the enzyme activity measurements was taken at similar times as the fibroblast biopsy?
The control used in Seahorse measurements was a fibroblast cell line routinely used in the Translational Metabolic Laboratory (Radboud University Medical Centre) for this porpoise. The skin biopsy was performed at 19 years old. Muscle biopsy was performed at 1 year old. This information was added to the manuscript article: line 99 of clinical profile and line 38 of cell culture.
Results line 176, 179, 181 – authors should add the appropriate references or web-links for the prediction tools used in the study.
References were added.
Results line 189 – Please delete the additional ‘c’ in cc.670G>A
Corrected.
Results lines 194-196 – I find the description of the results from the OCR and ECAR rates insufficient and the authors need to rewrite this section and provide more details of the results and discuss their findings.
In the discussion section, the interpretation of the results from the OCR and ECAR is extended “Mitochondrial function characterization using the patient’s fibroblasts revealed a mitochondrial dysfunction due to abnormal mitochondrial respiration, as evidenced by a decrease in the OCR/ECAR ratio. The decrease in OCR and the increase in lactate production observed in our patient are likely the consequences of deficient respiration due to general impairment of the entire set of mitochondrial respiratory chain complexes”.
Figure 1a – Parental sequencing chromatographs are shown but the sequencing for the proband is not shown. Can the authors provide the sequencing analysis for the affected patient?
Figure 1a was designed to show the segregation of the variants found in the proband, that is the reason it only shows the parents´ chromatograms. In our opinion, the sanger chromatograms of the proband don´t add information, but we could provide them to the reviewer if he/she finds it necessary.
Figure 2a – there is a spelling mistake in the y-axis for citrate synthase
Corrected.
Figure 2 – can the authors ‘tidy up’ the figures e.g. remove additional lines and align text so it’s not below the lines (figure 2a).
Done.
Reviewer 3 Report
The topic is interesting, to expand the clinical phenotype in mitochondrial disorders due to nuclear gene mutations is a very important task. There are increasing number of such cases thanks to the NGS technologies and these case studies enrich our knowledge about the mitochondrial medicine.
Suggestions:
The clinical description should be more detailed, beside the clinical symptoms the different laboratory investigations should be described in details.
- It should be described what was the best motor performance, presently there is only a half sentence indicates that the patient was not able to sit without assistance, but it is not clear if this was the best performance? Did the patient had ataxia as well, or the muscle paresis was the only reason of this bad performance. Did the patient had pyramidal signs?
- Regarding the laboratory investigations muscle histology should be described in details, not only the results of the COX and modified Gömöri Trichrome staining are interesting. Since it is a mitochondrial disease the electronmicroscopic results are also interesting for the readers.
- What about the results EMG and ENG
- The manuscript reports about one EEG, which was normal. It should be described at which age it was performed. Since the patient had a long follow up and several antiepileptic medication were introduced it would be interesting to know more about it.
The authors discuss the differences of the Complex activities in muscle and in fibroblasts, however in the Results section I could not find the results of the measurements from the muscle tissue
Conclusion: the manuscript reports about a clinically important topic, it is worth to publish it after improving the clinical details.
Author Response
We thank the reviewers for their positive comments and suggestions that have certainly helped to increase the overall quality of the manuscript. All the additions to the manuscript are highlighted in yellow in the revised version.
Suggestions:
The clinical description should be more detailed, beside the clinical symptoms the different laboratory investigations should be described in details.
Results from laboratory investigations were usually normal except for alkaline phosphatase (120 – range 30-106), pyruvate (0.70 – range 0.4-0.6), amino acids taurine (138 – range 20-90) and valine (330 – range 74-321). Variations of very long-chain fatty acids were also detected but were later normalized. In our opinion, this information does not add important details to the clinical profile, but if the reviewer feels this should be added to the manuscript we will include these details.
It should be described what was the best motor performance, presently there is only a half sentence indicates that the patient was not able to sit without assistance, but it is not clear if this was the best performance? Did the patient had ataxia as well, or the muscle paresis was the only reason of this bad performance. Did the patient had pyramidal signs?
Yes, it is the best achievement achieved by the patient and due to cerebellar atrophy, it is associated with the pyramidal syndrome, spasticity, and ataxia.
Lines 94-95 of the clinical profile section: "The patient required assistance to remain seated, which is her best motor performance achieved."
Regarding the laboratory investigations muscle histology should be described in details, not only the results of the COX and modified Gömöri Trichrome staining are interesting. Since it is a mitochondrial disease the electronmicroscopic results are also interesting for the readers.
We do not have the results of electronmicroscopic since muscle histology was performed a long time ago, and the technique was not available at the time at the hospital. Muscle biopsy showed the preserved architecture, with a uniform caliber of the fibers, without structural alterations of the fibers, and no ragged red fibers were observed.
Line 01 of clinical profile: “with uniform caliber of the fibers, without structural alterations”
What about the results EMG and ENG
EMG and ENG information were not available on the patient’s clinical history.
The manuscript reports about one EEG, which was normal. It should be described at which age it was performed. Since the patient had a long follow up and several antiepileptic medication were introduced it would be interesting to know more about it.
We agree with the reviewer's suggestion. The information was updated in the clinical profile, lines 93 and 94: “Electroencephalogram readings were initially normal, although results at 19 years old showed irregular and slightly slowed brain activity for age.”
The authors discuss the differences of the Complex activities in muscle and in fibroblasts, however in the Results section I could not find the results of the measurements from the muscle tissue
Muscle biopsy was not performed in our laboratory and we had access to the interpretation of the results from the patient’s clinical history. Although, we can provide them if the reviewer finds it necessary.
Conclusion: the manuscript reports about a clinically important topic, it is worth to publish it after improving the clinical details.
Reviewer 4 Report
The comments are in the attached file.

Author Response
We thank the reviewers for their positive comments and suggestions that have certainly helped to increase the overall quality of the manuscript. All the additions to the manuscript are highlighted in yellow in the revised version.
Major Comments
- Authors claim that they have identified a new variant, but they haven’t shown any protein expression or mRNA expression levels. I would recommend at least checking the protein and transcript level because authors have the patient’s fibroblast. The best solution would be to rescue the phenotype of the fibroblasts with wild type cDNA.
We agree with the reviewer that to demonstrate the functional link between variants and phenotypes should be the goal when describing new variants or new genes. On the other hand, the main aim of our work was to describe a new patient with a LTBL plus phenotype associated with EARS2 variants. The genotype of the patient, an already described missense variant and a new nonsense variant, fit perfectly with the molecular bases of the disease. The combination of severe change, like nonsense, with milder missense mutations is a common molecular mechanism in EARS2 associated diseases.
The EARS2 c.376C>T (p.Gln126Ter) is classified as pathogenic following ACMG rules, especially because it fulfils the PVS1 (pathogenic very strong) criteria: “ Null variant (nonsense, frameshift, canonical ±1 or 2 splice sites, initiation codon, single or multiexon deletion) in a gene where LOF is a known mechanism of disease”. So we think that from a clinical point of view the results are robust enough to establish a genetic diagnostic.
Minor Comments
- To ensure reliable clinical interpretation of the detected variants, prioritization criteria were applied to predict their pathogenicity in accordance with AMCG guidelines. Please correct the typo: American College of Medical Genetics and Genomics (ACMG) (page no-3; line-131-132).
Corrected.
Please read carefully the manuscript and correct typo (example: The assay mix contained 0,3 mM acetyl-CoA, 0,1 mM DTNB, 0,025% Triton 161 X-100, 10mM Tris-HCl pH 8,1. Figure1 legend cc.670G>A (p.Gly224Ser))
Corrected.
- Please italicize EARS2 (EARS2) when indicating the gene. It is inconsistent throughout the manuscript. Please make sure that punctuations are applied properly.
Corrected.
- Please make proper citations of the software used for population frequencies, conservation scores, and functional prediction (example- GERP, PhyloP, SIFT, Provean, MutationTaster).
References were added.
- Please follow the nomenclature recommended by the Human Genome Variation Society (HGVS) in figure-1b. Check the nomenclature within the red boxes. There are some inconsistencies.
Following reviewer suggestions, this was corrected.
Please notify in Figure 1b the variants detected in this case with either box, bold or red characters etc.
Following reviewer suggestions, we highlighted the variants detected in our case in bold.
- There is a typo on the Y-axis of figure-2a. It is redundant to write chart title above the histogram (Ex-OCR/ECAR) since Y-axis already described that. There is no statistical analysis in figure 2a and 2b. Please do statistical analysis and show if it is significant.
Typo on the Y-axis of figure 2a corrected and titles above both charts were deleted.
Mitochondrial stress tests performed with Seahorse Bioscience XF Analyzers allow the estimation of different bioenergetic measures by monitoring the oxygen consumption rates of fibroblasts in multi-well plates in the same experiment. Normalization of the resultant data minimizes inconsistency and variations from well-to-well, average, and the standard deviation is then calculated. Statistical analysis is not usually performed in these tests. Only the Grubbs’s test is applied for the identification of outliers.
- In figure 3, capital letters (A, B) were used, whereas in figure 1 and 2, small letters (a, b) were used. Please follow the journal’s instruction.
Corrected.
- Authors claim that they have identified a new variant, but they haven’t shown any protein expression or mRNA expression levels. I would recommend checking the protein or transcript level because authors have the patient’s fibroblast.
Previously answered in the major comment.
- Please apply correct punctuations in the abbreviation section of the supplementary tables 1 and 2. Punctuations were not correctly used. In supp table-1, symbol(†) was used instead of the symbol (*), which represents stop codon.
Corrected.
- Please check the references if all are cited in the text.
Manuscript and supplementary material references were updated.
Round 2
Reviewer 1 Report
The questions 1, 4, 5, 6, 7 and 8 were not adequately addressed. I regret that I do not feel the manuscript has substantially improved in this round of revision.
Author Response
We thank the reviewer for the comments that indeed helped to improve the quality of this manuscript. All the additions to the manuscript are highlighted in green in the second round of the revised version.
Comments and suggestions:
1. While I understand using a set of targeted panel, the authors should indicate why they excluded analyzing the mtDNA from the patient, which can contribute/explain the variation in the clinical presentations.
We try to clarify this point detailing that patient´s mtDNA was sequenced at two years old according to the clinical records, and no pathogenic variants were identified. We agree this should be clarified and we have added the sentence in line 103 of the manuscript, in the clinical profile section.
“Patient´s mtDNA was sequenced and the presence of pathogenic variants were discarded”
Although the suggestion of the reviewer about the role of mitochondrial DNA as a factor behind the variation in the clinical presentations associated with EARS2 variants is very interesting, we think this would imply a new study in which we would need the mtDNA sequence not only of our patient, but also the sequences of the reported LTBL patients.
4. The measurement of OCAR are not always normalized by CS, although it is fine. The authors should be clear in their interpretation of what it means when the data are normalized this way, i.e. OxPhos activity per mitochondria, which does not necessarily correlate to OxPhos activity per cell.
In the previous round of revisions, we misunderstood the reviewer comment. We agree that this needed to be clarified so we have made the following changes in the manuscript:
- We have changed the sentence “OCR was measured before and after the addition of inhibitors and were normalized to the activity of citrate synthase (CS) [22] to avoid the effect of differences in mitochondrial content or cell number.” Lines 170-171 of Experimental Section.
- We have changed the sentence in lines 205-206 of results section to clarify that OCR was normalized to mitochondrial content: “Measurement of the OCR and extracellular acidification rate (ECAR), normalized by mitochondrial content,”
5. The most troubling aspect of the report is the lack of direct verification of suspected mitochondrial translation deficiency. The authors should demonstrate with western blot analysis or immunofluorescence analysis that the compound heterozygous EARS2 variants reported here lead a breakdown of OxPhos complexes. Alternatively, the mitochondria translational rate should be measured to demonstrate the casual relationship.
We agree with the reviewer’s comment in which the demonstration of the molecular consequences of new variants is the gold standard assigning pathogenicity. Unfortunately to perform those experiments is not always possible and in our case, the experimental part of the study was developed during an internship in Radboud Hospital. At this moment to continue with the molecular characterization of the new variant will deeply delay the publication of the manuscript.
On the other hand, we think that from a clinical point of view the results are robust enough to establish a genetic diagnostic, which was the main goal of this study.
Previous answer:
We agree with the reviewer that to demonstrate the functional link between variants and phenotypes should be the goal when describing new variants or new genes. On the other hand, the main aim of our work was to describe a new patient with a LTBL plus phenotype associated with EARS2 variants. The genotype of the patient, an already described missense variant, and a new nonsense variant, fit perfectly with the molecular bases of the disease. As we have answered before, the combination of severe change, like nonsense, with milder missense mutations is a common molecular mechanism in EARS2 associated diseases. So we think that from a clinical point of view the results are robust enough to establish a genetic diagnostic.
6. It would be fairly easy to examine how the state of OxPhos complex assembly might impact the health of mitochondria population from the patient fibroblasts. Are mitochondria mass and morphology altered? Again, simple staining of mitotracker would answer this question. Also, are the ROS burden of these cells altered, which can be measured using specific dye? If the mitochondria become more depleted due to the reported EARS2 variants, then one can expect the mitochondria dysfunction to be even more enhanced when compared on the cellular level.
We agree that this should be clarified, and we have added a sentence in line 212-213 of the results section pointing out the differences in mitochondrial mass between patient and control fibroblasts: “Also, mitochondrial mass differences were detected between patient and control, 160 U/l and 339 U/l, respectively, regarding to CS measurement.”
As we tried to answer previously, the study of ROS metabolism related to the pathogenesis of EARS2-disorders would be indeed of great interest, but we didn´t performed that cellular characterization.
7. For the Seahorse measurements, what control is used? This is not clearly described. Although it might be difficult, it would be interesting to compare to the samples from the parents, especially the novel variant, which presumably has only subtle effects.
We have added a sentence in lines 140-141 explaining the use of a fibroblast cell line routinely used for research porpoises.
8. How were the measurements in Fig 2c carried out? Is this done with purified mitochondria or mitochondria extracts? It is unclear why the method section listed the method to purify mitochondria. The reference cited offered a range of methods and the author should explicitly explain which methods were used. Or the authors should state what company manufactured the kit if a kit were used. The data from this experiments was indeed surprising. Is there compensating mechanism at work here?
We have changed the enzymatic activities assays paragraph (lines 173-176) in the experimental section to try to answer the reviewer comments: “The enzymatic activities of complexes I–V were assayed spectrophotometrically using mitochondrial extracts prepared as previously described in mitochondrial isolation. These measurements were performed according to the current protocols in use in the Nijmegen Center for mitochondrial disorders.”
The differences between muscle and fibroblast OXPHOS activities could be surprising but is a common finding in aminoacyl-tRNA synthetase associated diseases. That point is discussed in lines 281-283 of the discussion section, although the mechanism behind this phenomenon is unclear.
Reviewer 2 Report
n/a
Author Response
We thank the reviewer for the suggestion that indeed helped to improve the quality of this manuscript. According to the suggestion, English language was reviewed and spell was checked.
All the corrections to the manuscript are highlighted in green in the second round of the revised version.